# Exposure to *Brucella* spp. in Goats and Sheep in Karenga District, Uganda Diagnosed by Modified Rose Bengal Method

Claire Julie Akwongo [1,*] and Steven Kakooza [1,2]

1  Vétérinaires Sans Frontières Germany, Kampala P.O. Box 24384, Uganda
2  Central Diagnostic Laboratory, College of Veterinary Medicine, Animal Resources and Biosecurity, Makerere University, Kampala P.O. Box 7062, Uganda
*  Correspondence: juliemanifest@gmail.com

**Simple Summary:** Brucellosis is a disease of public health importance and with great impacts on livestock production and food security. Controlling the disease requires a combined effort from both animal and human health sectors, and one of the steps towards achieving this is through understanding how widespread it is and the factors influencing the spread, so as to inform decisions on the prioritization of control measures, resource allocation, and risk communication. We therefore believe that our study will add new information on the epidemiology of brucellosis in Uganda, particularly in the Karenga district, where data on brucellosis in small ruminants has never been published.

**Abstract:** A retrospective study was conducted in 2022 to determine the seroprevalence of brucellosis and its risk factors among goat and sheep herds in the Karenga district, Karamoja region, Uganda. Sera from 332 goats and 81 sheep from 20 kraals in all seven administrative units of the district were tested using the modified Rose Bengal test (mRBT). It was determined that brucellosis was present in 20% of the small ruminant herds in the Karenga district, with an overall animal level seropositivity of 3.39% (CI: 1.87–5.62%; n = 14/413). Higher seropositivity was recorded in goats (3.92%, CI: 2.1–6.6%) than sheep (1.23%, CI: 0.03–6.69%). Similarly, seropositivity was higher in females (3.95%, CI: 2.12–6.66%) than males (1.19%, CI: 0.03–6.456), and also higher in goats and sheep more than two years old (4.15%, CI: 2.09–7.31%) than those one to two years old (1.98%, CI: 0.24–6.97%) or those less than one year old (2.13%, CI: 0.05–11.29%). Only herd size was statistically significant ($p < 0.05$) as a risk factor for exposure to brucellosis in the Karenga district. Although the prevalence of brucellosis among goats and sheep in the Karenga district is low, there still exists a risk to the pastoralists who closely interact with the animals and also practice risky behaviours such as consumption of uncooked or unpasteurised goat milk. There is need to expand herd health messaging in Karenga to include risk communication on brucellosis and other zoonotic diseases.

**Keywords:** brucellosis; goats; sheep; One Health; zoonoses; pastoralists





## 1. Introduction

Brucellosis is an important zoonotic disease caused by an intracellular Gram-negative bacteria of the genus *Brucella* [1,2], with the species of concern being *B. abortus* in cattle, small ruminants, bison, and cervids; *B. suis* in swine, reindeer, cattle, and bison; and *B. melitensis* in small ruminants. Other species *B. canis* and *B. ovis* affect dogs and sheep, respectively [3]. The bacteria in animals primarily localize in the reproductive organs and/or the udder, and are shed in milk or via aborted foetuses, afterbirth, or other reproductive tract discharges. Contact with the contaminated formites facilitates pathogen transmission from infected hosts to susceptible animals and humans [3]. Infected animals may not appear to be clinically sick but may manifest an apparent reduction in fertility characterised by poor conception rates, retained afterbirths with resulting uterine infections, and enlarged, arthritic joints [4]. Pregnant animals usually abort in the last trimester or

give birth to weak offspring. Changes in the normal lactation period as a result of abortions and/or delayed conceptions can lead to a reduction in milk production [4,5].

Humans contract brucellosis through consumption of contaminated animal products such as undercooked meat, unpasteurised dairy products, such as milk, fresh cheese, cream, ice cream, or yoghurt, and also through use of other animal products including hides and skins which are not well-aged [6,7]. The disease in humans often presents with non-specific clinical signs similar to those seen in other febrile diseases such as fever, chills, headaches, night sweats, muscle aches, joint pain, weakness, weight loss, nausea, and depression, and thus may often be misdiagnosed [8,9]. Wrong or delayed treatment of infected persons allows for the development of further complications that may involve the heart and nervous system, persistent long-term joint and bone disorders, and death [8].

In East Africa, animal-level prevalence ranges of 0.2% to 43.8%, 0.0% to 20.0%, and 0.0% to 13.8% are estimated for cattle, goats, and sheep, respectively, and between 0.0% and 35.8% for humans [10]. On the other hand, in Uganda, human and animal brucellosis prevalences have been projected at 10% and 5.5%, respectively [11]. Most of the studies in Uganda have, however, been performed on cattle populations, with very few on small ruminants. As *Brucella melitensis* and *Brucella abortus* are usually implicated in human brucellosis cases, it is important to control the disease in goats and sheep [3,12]. This requires an understanding of the epidemiological situation including prevalence in the small ruminant populations. This information from the Karenga district has never been reported before, even though small ruminants keeping is a major source of livelihood and provides the main source of meat and milk for the pastoralists. The present study therefore sought to screen goats and sheep in the Karenga district for brucellosis and establish the epidemiological situation regarding the disease in the district. This information will be useful for policymakers and technical personnel in animal and human health to evaluate the disease status in livestock and develop strategies to prevent transmission to humans, in the spirit of One Health.

## 2. Materials and Methods

### 2.1. Study Location

A retrospective study was performed using samples from goats and sheep, collected from the Karenga district. The district is located in North Eastern Uganda in the Karamoja region, and is bordered by South Sudan to the northwest, Kaabong district in the east and south, Kotido district in the south, and Kitgum district in the west (Figure 1). The district is inhabited by the Karimojong, Acholi, and Mening tribes, for whom livestock keeping constitutes the major source of livelihood. Cattle, goats, sheep, and donkeys are kept under the transhumance system characterised by seasonal movement of people and their livestock from one place to another in search of water and pasture [13]. Like other districts in the Karamoja region, Karenga district is semi-arid but distinct wet and dry seasons are a prominent feature, with the average rainfall at about 519 mm per annum. The daily temperatures range between 20 °C and 32 °C and relative humidity can reach 60% [13,14].

### 2.2. Study Samples and Data

The study analysed 413 serum samples from unvaccinated goats (332) and sheep (81), randomly selected from a batch of samples that were previously collected for a study by [15], which employed a two-stage cluster sampling approach to select 22 kraals from all seven administrative units of the Karenga district, from which a total of 684 goats and sheep were selected by systematic random sampling, between November and December, 2020. The current study selected serum samples from the batch of 648 that had been stored in a freezer at −30 °C at the Central Diagnostic Laboratory, Makerere University. Corresponding animal biodata that accompanied the samples were used to perform analyses on potential risk factors.

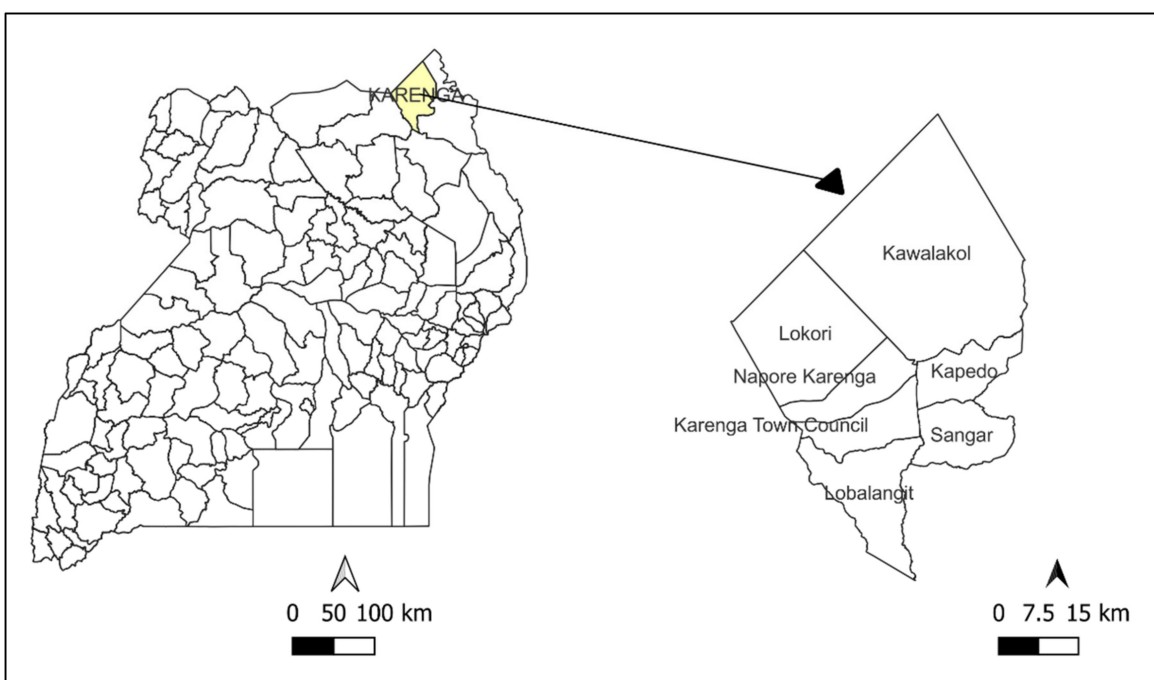

**Figure 1.** Map of Uganda showing location of the Karenga district. Map was developed by the primary author in QGIS Desktop 3.22.9 using shape files from UNHCR [(https://data.unhcr.org/en/documents/details/83043) (accessed on 7 August 2022)] and Pearl Geo portal [(https://pearlgeoportal.com/district/Karenga/shapefiles) (accessed on 7 August 2022)].

*2.3. Laboratory Analysis*

The laboratory analysis was conducted at the Central Diagnostic Laboratory at Makerere University, using the modified Rose Bengal test (mRBT) described in a previous study [16]. Briefly, the screening commercial antigen (Rose Bengal rapid slide agglutination antigen kit, RSA-RB from IDVet, Grabels, France) was brought to room temperature and gently shaken to obtain a homogenous suspension. The test sera were also brought to room temperature 30 min before testing. Seventy-five (75 µL) of the test sera and 25 µL of the antigen were placed side by side on a white tile and mixed thoroughly, after which the mixture was rocked for 4 min before reading the results. Plate reading was done immediately under good lighting, observing with the naked eye for any agglutination. Test results were interpreted as positive (if there was any agglutination seen) or negative (if there was no agglutination). For quality control, known positive and negative control sera were used to validate the reagent and procedures done.

*2.4. Data Analysis*

Study data were collated and managed in Microsoft Excel version 2016 and descriptive analysis was conducted using the R console version 4.2.0, with a 95% confidence interval (CI) and 5% significance level. Seroprevalence was computed by dividing the number of positive animals by the total number tested, and herd/flock seroprevalence was calculated by dividing the number of flocks/herds with at least one test positive animal by the total number of herds/flocks tested. The corresponding prevalence CIs were computed as exact binomial 95% CIs. Statistical significance of association between the individual animal seropositivity for *Brucella* antibodies and the potential risk factors including age, sex, location, animal species, herd structure, herd/flock size, and production system were determined by univariate logistic regression. Other potential risk factors including communal grazing and watering of livestock and sharing breeding bucks/rams were not analysed because they were practiced in all kraals. Significant variables from the univariate logistic regression analysis (with $p < 0.05$) were further analysed in a multivariable logistic

regression model, employing a forward and backward stepwise Akaike Information Criteria (AIC) to identify significant risk factors for *Brucella* seropositivity in the Karenga district. Regression coefficients were converted to odds ratios and their confidence intervals and goodness of fit of the final model was tested using the Hosmer Lemeshow test.

## 3. Results

### 3.1. Animal Characteristics

Serum samples from a total of 332 indigenous small East African goats and 81 black-headed Persian sheep were included in the study. Of the 332 goats, a higher proportion were female (n = 256, 77.1%) compared to male (n = 76, 22.9%). Similarly, out of the 81 sheep, there were more females (n = 73, 90.1%) than males (n = 8, 9.9%). Most of the animals were older than two years of age (n = 265), followed by one to two years old (n = 101), while a few were less than a year old (n = 47).

### 3.2. Brucella Seropositivity

Out of the 413 serum samples tested, 14 (3.39%; 95% CI: 1.87–5.62%) were positive for brucellosis. Species-aggregated prevalence for goats and sheep were 3.92% (CI: 2.10–6.60%) and 1.23% (CI: 0.03–6.69%), respectively, while herd-level seropositivity was 20% (Table 1).

**Table 1.** Prevalence estimates of brucellosis among goats and sheep in the Karenga district, Uganda by modified Rose Bengal Test.

| Animal Species | Number Tested | Number Positive | Proportion of Seropositive Animals (%) |
|---|---|---|---|
| Ovine | 81 | 1 | 1.23 |
| Caprine | 332 | 13 | 3.92 |
| Total | 413 | 14 | 3.39 |

### 3.3. Univariate Analysis of Probable Risk Factors for Brucella Seropositivity in the Karenga District

Out of the six variables analysed in the univariate analysis for seroprevalence of brucellosis in the Karenga district, five showed a significant association ($p < 0.05$) with brucellosis seropositivity.

As shown in Table 2, brucellosis was detected in sheep and goats in five out of the seven administrative units in Karenga district, with the Kapedo subcounty registering the highest seroprevalence (11.11%, CI: 3.11–26.06%). Seroprevalence was higher in goats (3.92%, CI: 2.1–6.6%) than in sheep (1.23%, CI: 0.03–6.69%), and was higher in females (3.95%, CI: 2.12–6.66%) than males (1.19%, CI: 0.03–6.456%). Exposure was higher (4.15%, CI: 2.09–7.31%) in goats and sheep more than two years old than those one to two years old (1.98%, CI: 0.24–6.97%) or those less than one year old (2.13%, CI: 0.05–11.29%).

**Table 2.** Univariate Logistic Regression analysis of risk factors for Brucella seropositivity among goats and sheep in the Karenga district, Uganda.

| Variable | Number Tested | Number Positive | Prevalence (%) | 95% CI | *p*-Value |
|---|---|---|---|---|---|
| **Animal Species** | | | | | $\mathbf{1.33 \times 10^{-5}}$ *** |
| Ovine | 81 | 1 | 1.23 | 0.03–6.69 | Ref |
| Caprine | 332 | 13 | 3.92 | 2.10–6.60 | 0.258 |
| **Age category** | | | | | $\mathbf{4.66 \times 10^{-8}}$ *** |
| <1year | 47 | 1 | 2.13 | 0.05–11.29 | 0.953 |
| 1–2 years | 101 | 2 | 1.98 | 0.24–6.97 | Ref |
| >2 years | 265 | 11 | 4.15 | 2.09–7.31 | 0.327 |

**Table 2.** *Cont.*

| Variable | Number Tested | Number Positive | Prevalence (%) | 95% CI | *p*-Value |
|---|---|---|---|---|---|
| **Sex** | | | | | $1.12 \times 10^{-5}$ *** |
| Male | 84 | 1 | 1.19 | 0.03–6.456 | Ref |
| Female | 329 | 13 | 3.95 | 2.12–6.66 | 0.24 |
| **Herd structure** | | | | | $2.33 \times 10^{-14}$ *** |
| Goat only | 193 | 4 | 2.07 | 0.57–5.22 | Ref |
| Mixed herd | 220 | 10 | 4.55 | 2.20–8.20 | 0.177 |
| **Herd size** | | | | | $2.92 \times 10^{-7}$ *** |
| Medium | 172 | 1 | 0.58 | 0.02–3.19 | Ref |
| Large | 241 | 13 | 5.39 | 2.90–9.05 | 0.0289 * |
| **Location** | | | | | 0.995 |
| Karenga Town Council | 12 | 0 | 0 | 0.00–2.65 | Ref |
| Lobalangit | 79 | 1 | 1.27 | 0.03–6.85 | 0.996 |
| Karenga Subcounty | 71 | 1 | 1.41 | 0.04–7.59 | 0.996 |
| Kapedo | 36 | 4 | 11.11 | 3.11–26.06 | 0.996 |
| Kawalakol | 90 | 6 | 6.67 | 2.49–13.95 | 0.996 |
| Sangar | 45 | 2 | 4.44 | 0.54–15.15 | 0.996 |
| Lokori | 80 | 0 | 0.00 | 0.00–4.51 | 1 |

CI-Confidence interval; Ref-Reference category; *, ***-Statistically significant.

### 3.4. Multivariate Analysis of Risk Factors for Brucella Seropositivity in the Karenga District

The final multivariable logistic regression model (Hosmer Lemeshow test, *p* = 0.9112) contained three variables including animal species, sex, and herd size, out of which only herd size was statistically significant, with the odds of being positive for brucellosis being 3.3 times higher in large herds than smaller herds/flocks (Table 3).

**Table 3.** Multivariable logistic regression analysis of risk factors for Brucella seropositivity in the Karenga district.

| Variable | Odds Ratio | Std. Error | 95% CI | z Value | *p*-Value |
|---|---|---|---|---|---|
| **Animal Species** | | | | | |
| Ovine | | | | | Ref |
| Caprine | 6.01 | 1.052 | 1.1–8.2 | 1.705 | 0.0881 |
| **Sex** | | | | | |
| Male | | | | | Ref |
| Female | 3.6 | 1.053 | 2.3–17.9 | 1.208 | 0.2270 |
| **Herd size** | | | | | |
| Medium herd (50–100) | | | | | Ref |
| Large herd (more than 100) | 3.3 | 1.046 | 2.3–16.4 | 2.383 | 0.0172 * |

CI-Confidence interval; Ref- Reference; *-Statistically significant.

## 4. Discussion

Brucellosis is endemic in Uganda's society, impacting the health of humans and animals, ultimately affecting the national economy. On 3 November 2016, the Government of Uganda launched a National One Health Platform (NOHP) to forefront synergistic efforts amongst various health sector actors in the fight against existing zoonotic diseases. To tackle zoonotic disease challenges, a key activity was to prioritize diseases of national public health concern based on contributions from representatives of human health, veterinary, environment, and wildlife sectors. The ranking listed the following priority diseases

for Uganda: anthrax, zoonotic influenza viruses, viral haemorrhagic fevers, brucellosis, trypanosomiasis (African sleeping sickness), plague, and rabies [17]. Therefore, these results are valuable to support One Health brucellosis control efforts at disease control frontline ministries such as the Ministry of Health (MOH) and Ministry of Agriculture, Animal Industry and Fisheries (MAAIF) in Uganda.

The European Union considers that the best approach for diagnosing sheep and goat brucellosis is the mutual use of Rose Bengal test (RBT) as the screening method and the complement fixation test (CFT) as the confirmatory test [18]. It has been reported that high numbers of sheep and goats from B. melitensis-infected flocks show negative results when tested with the standard RBT but show positive results from the CFT, thus questioning the sensitivity of the RBT as a screening test [4]. Thus, the concurrent use of both is recommended to obtain a maximal sensitivity [4,12]. To improve the sensitivity of the RBT, a simple modification that involves increasing the volume of sera to be tested has been recommended [4,5,19]. The mRBT significantly increases the sensitivity of the standard RBT procedure and greatly reduces the problem of sera being RBT negative but CFT positive [16].

The current study established the presence of brucellosis among small ruminants in the Karenga district and reported that 20% of the herds/flocks were positive for brucellosis, with animal level prevalence being 3.39%. Despite the relatively low prevalence, this could pose a high risk to the human population in the Karenga district as the culture of consuming raw milk, sometimes directly from the udder, is prevalent among the pastoralists in the area.

The prevalence reported in this study is higher than the 1.2% in cattle and 0.3% in goats reported by [20] in the Iganga district but lower than the 8.85 % in sheep and 10.52% in goats reported by Dubad, et al. [21] in Kiruhura. In a previous study by Nyerere, et al. [22], climatic factors played a role in the transmission dynamics of brucellosis in humans, livestock, and wild animals, which we anticipate could also contribute to the variations in brucellosis prevalence in the Karenga, Iganga, and Kiruhura districts in Uganda. The prevalence findings in the Karenga district are also below the average regional and national prevalence of brucellosis, which ranges between 4% and 10%, by various serological tests [23,24]. While there are differences in *Brucella* seroprevalences in small ruminants across Uganda, the figures are much lower than those reported in other parts of the world, for example, Hawari [25] reported *Brucella* prevalence of 21.1% and 24.6% in sheep and goats respectively in West Bank, Jordan. With these low figures, if all efforts are placed on surveillance and utilization of existing control measures such as vaccination, the disease can easily be eradicated from Uganda.

Larger herds were significantly more exposed to brucellosis than smaller herds in the Karenga district. Similar findings were reported by Mugizi, et al. [26] in the Gulu and Soroti districts among cattle herds, by Makita, et al. [27] in Kampala, by Langoni, et al. [28] in Brazil, and by Mekonnen, et al. [29] in Western Tigray, Ethiopia. Close contact and interaction between animals in large herds could facilitate the spread of brucellosis from infected to susceptible members of the herd, as compared to smaller herds. This can be further exacerbated by mixing of different animal species in the herd. Cross-transmission of brucellosis has been reported to occur among cattle, sheep, goats, camels, and other species [30].

In this study, even though not statistically significant, mixed herds recorded more positive animals than single-species herds/flocks. Studies by Kabagambe, et al. [31] in eastern and western Uganda identified mixing of sheep and goats as being a risk factor for *Brucella* positivity. Brucellosis prevalence was higher in goats than sheep, but the difference was not statistically significant. Dubad, Baluka, and Kaneene [21] also reported higher prevalence in goats than sheep in the Kiruhura district in Uganda. A higher seroprevalence in goats than in sheep has also been described in other countries including Ethiopia [32], Iran [33], and Kosovo [34]. However, higher prevalences were reported in sheep than goats in Burkina Faso [35] and India [36]. These variations in results from different studies can be attributed to different management systems, study designs, and sample sizes.

The higher seroprevalence in females than males reported in this study is consistent with reports in Bangladesh by Shafy, et al. [37] and in Uganda by Dubad, Baluka, and Kaneene [21]. This difference could be explained by the fact that the absence in males of erythritol (found in the gravid placenta), where the bacteria multiplies, makes them less susceptible to brucellosis [38]. Additionally, like for other livestock diseases as reported by [15], the fact that females are kept longer in the herd for breeding purposes could explain the higher prevalence rates in females than males as they have more exposure time in the herds.

Additionally, even though location was not a statistically significant risk factor in this study, areas nearest to the Kidepo Valley National Park including the Kawalakol, Kapedo, and Sangar subcounties recorded the highest prevalences of brucellosis. In a previous study by Aruho, et al. [39], a significantly higher percentage (55.9%) of *Brucella*-positive samples from wildlife was reported in wildlife from the Kidepo Valley National Park than other national parks in Uganda. We hypothesize that there could be a possible spill over of infections from wildlife to domestic animals in these subcounties as livestock frequently graze together with wildlife in the Karenga district at these interfaces. Bugeza, et al. [40] reported sharing of water sources with wildlife as a risk factor for brucellosis in the Nakasongola district in Uganda.

## 5. Study Limitations

The Rose Bengal Test is one of the screening tests for brucellosis recommended by the World Organization for Animal Health. However, results from the screening test need to be confirmed by other recommended tests for this purpose.

## 6. Conclusions and Recommendations

Our findings confirm the exposure of goats and sheep to *Brucella* spp. and provide prevalence estimates for brucellosis in sheep and goat herds in the Karenga district based on the modified Rose Bengal screening test. There is need to confirm the results by other confirmatory tests, as well as characterise the *Brucella* spp. present in this population to provide a clear picture of the brucellosis epidemiological situation in the Karenga district. Despite the low prevalence values, brucellosis could possibly pose a public health threat due to the presence of practices such as consumption of unpasteurised/raw milk by pastoralists and the absence of control programmes in the district (as existing herd health programmes do not cover brucellosis). There is need to expand the herd health programmes to include risk communication on the disease and further steps to include vaccination against *Brucella* spp. could go a long way in eradicating the disease from the population, considering the low prevalence values. Evaluation of brucellosis in other species including humans is necessary to provide a full picture of the disease epidemiology in the Karenga district and facilitate a comprehensive control and prevention programme.

**Author Contributions:** C.J.A. contributed to the study design, statistical analysis and drafting of the manuscript. S.K. contributed to the study design and performed the laboratory analysis of samples. All authors have read and agreed to the published version of the manuscript.

**Funding:** This research received no external funding.

**Institutional Review Board Statement:** No ethical approval was required as the study did not collect primary data and samples from animals or their owners but used already existing serum samples that were stored in the laboratory. Access to the samples was approved by the researcher who collected them.

**Informed Consent Statement:** Not applicable.

**Data Availability Statement:** All data relating to this study are available on request from the corresponding author.

**Acknowledgments:** The authors acknowledge the management and staff of Central Diagnostic Laboratory, Makerere University for providing laboratory space for sample storage and analysis.

**Conflicts of Interest:** The authors declare no conflict of interest.

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
