# Peer review of "Exposure to Brucella spp. in Goats and Sheep in Karenga District, Uganda Diagnosed by Modified Rose Bengal Method"

_zoonoticdis, doi:10.3390/zoonoticdis2030015_

Round 1
Reviewer 1 Report
The manuscript studies the epidemiological situation regarding brucellosis in the Karenga district, which information will be useful for policymakers and technical personnel in animal and human health to evaluate the disease status in livestock and develop strategies to prevent transmission to humans, in the spirit of One Health. The manuscript needs a lot of improvement. Here are some recommendations;
- The manuscript needs to be revised by a native English speaker.
- Add the Ethical statement in the methods section with the protocol number.
- The data presented in the manuscript is not enough and you will need more experiments to complete the story. Presenting the prevalence data is not enough to make a complete story from the collected samples. You need to use sequencing to characterize the positive samples, Also, what about the confirmatory tests.
Reviewer 2 Report
This manuscript adds to our epidemiological understanding of Brucellosis prevalence rates in a district of Uganda. Therefore, the contribution is incremental but provides useful Methodologies used are
Some recommended edits are as follows:
--Consider providing a graphic/map of the surveyed region
--More detail is needed about the sampling strategy in order to understand the epidemiologic relevance of the data. Was the sampling random? What was the basis for collection of the samples? I feel like this addition is of particular importance for this paper.
--Ideally provide a brief summary of the collection and processing procedures for the serum samples themselves.
--Include in the discussion of data the relevance and significance of the particular test used for detection (modified Rose-Bengal). Also comment on how the test type used can influence results as compared to other studies that may use different diagnostic approaches.
--In terms of the odds for positivity in large herds being smaller than the odds for small herds--please comment on whether this has anything to do with other confounding variables such as sampling strategy.
Reviewer 3 Report
all is in att file

Reviewer 4 Report
The study highlights an important topic on brucellosis, which is significant from a 'One Health' standpoint as it is a zoonotic disease and affects both animals and human beings. The manuscript is well written and flow logically. The analysis of results is extensive. However, the study design is fundamentally flawed in a number of respects:
i. The Rose Bengal test alone cannot be used to determine sero-prevalence. Even if the authors stated the limitations of using the Rose Bengal test, this still falls short because the authors should have further analysed the samples using tests such as SAT, CFT, FPA and/or ELISA as stated in the OIE Terrestrial Manual on diagnosis of brucellosis. For this reason, the findings of this study fall short of the minimum expected standard. Furthermore, the authors did not include reference sera for the RBT, which is a crucial aspect of this test. For this reason, the conclusion from the study may be inaccurate.
Minor comments: The authors should include the actual numbers of positive samples in the abstract.
Round 2
Reviewer 1 Report
The authors addressed my comments, However, there are still some typo errors, please correct
Author Response
Thank you for acknowledging the improvements to this paper based on your previous comments. We have taken time to review the manuscript further and corrected the typo errors noted with the spelling checksReviewer 2 Report
Overall, in their revision, the authors have generally addressed the comments from review in a suitable manner. A description of the context of the test used (modified Rose-Bengal) was added to the discussion section, and then this was paired with a discussion of the potential limitations of the use of this test in the absence of a confirmatory test. A map of the region sampled was also added.
The addition of a reference/link to the description of the study from which the samples were obtained was also important in terms of addressing review comments. However, I would recommend the addition of further text providing a summary of the means by which the original samples were collected. In other words--the samples used for this study were randomly selected from the pool of available samples. But how were the herds selected for the initial referenced study? I understand that information is contained in the referenced paper, but due to the importance of this information to the interpretation of the generalizability of this study, I think it should be summarized in this paper as well.
Author Response
Thank you for acknowledging the changes made to the manuscript based on your previous comments.
A sentence explaining sampling procedure from the reference study whose samples were used in this study has been added in line 96-105, section 2.2 of the revised manuscript.
Reviewer 4 Report
I would like to thank the authors for the responses to comments. I am of the view that given the well-documented sensitivity and specificity levels of the Rose Bengal Test (RBT); and that it is standard practise to confirm the results of RBT using additional tests such as the CFT or FPA, the manuscript does not meet the fundamental standard of the journal. If possible, the authors are strongly advised to perform the additional basic confirmatory tests. Otherwise, the article may be considered as a short communication.
Author Response
Thank you for acknowledging our response to your previous comments. Performing the CFT and FPA in our local situation is quite expensive and requires investment of huge resources which at hand we do not have, despite the fact that we would have loved to do more analyses.